# CinA mediates multidrug tolerance in *Mycobacterium tuberculosis*

Kaj M. Kreutzfeldt[1], Robert S. Jansen [2,5], Travis E. Hartman[2], Alexandre Gouzy [1], Ruojun Wang[1,6], Inna V. Krieger [3], Matthew D. Zimmerman[4], Martin Gengenbacher[4], Jansy P. Sarathy[4], Min Xie[4], Véronique Dartois [4], James C. Sacchettini [3], Kyu Y. Rhee [2,7], Dirk Schnappinger [1✉] & Sabine Ehrt [1✉]

The ability of *Mycobacterium tuberculosis* (*Mtb*) to resist and tolerate antibiotics complicates the development of improved tuberculosis (TB) chemotherapies. Here we define the *Mtb* protein CinA as a major determinant of drug tolerance and as a potential target to shorten TB chemotherapy. By reducing the fraction of drug-tolerant persisters, genetic inactivation of *cinA* accelerated killing of *Mtb* by four antibiotics in clinical use: isoniazid, ethionamide, delamanid and pretomanid. *Mtb* Δ*cinA* was killed rapidly in conditions known to impede the efficacy of isoniazid, such as during nutrient starvation, during persistence in a caseum mimetic, in activated macrophages and during chronic mouse infection. Deletion of CinA also increased in vivo killing of *Mtb* by BPaL, a combination of pretomanid, bedaquiline and linezolid that is used to treat highly drug-resistant TB. Genetic and drug metabolism studies suggest that CinA mediates drug tolerance via cleavage of NAD-drug adducts.

[1] Department of Microbiology and Immunology, Weill Cornell Medical College, New York, NY 10065, USA. [2] Division of Infectious Diseases, Department of Medicine, Weill Cornell Medical College, New York, NY 10065, USA. [3] Department of Biochemistry and Biophysics, Texas A&M University, College Station, TX 77843, USA. [4] Center for Discovery and Innovation, Hackensack Meridian Health, Nutley, NJ 07110, USA. [5] Present address: Department of Microbiology, Radboud University, 6525 AJ Nijmegen, The Netherlands. [6] Present address: Department of Molecular Biology, Princeton University, Princeton, NJ 08540, USA. [7] Present address: Department of Microbiology and Immunology, Weill Cornell Medical College, New York, NY 10065, USA. ✉email: dis2003@med.cornell.edu; sae2004@med.cornell.edu

Killing 1.5 million people last year, *Mycobacterium tuberculosis* (*Mtb*) remains the leading cause of death by a single infectious bacterium[1]. Partly responsible for the lack of success in fighting tuberculosis (TB) is the length and complexity of the required drug regimen, which still consists of at least 6 months of combination therapy with up to four drugs when TB is caused by drug-sensitive *Mtb*[2]. The need for such lengthy and complicated regimens is generally believed to be due to *Mtb* subpopulations that persist in the face of drugs that sterilize actively replicating *Mtb* in vitro. The mechanisms that enable *Mtb* to persist are complex. They include poor penetration of drugs into some infected lesions[2,3] and alterations in bacterial physiology leading to drug tolerance, a phenomenon that is distinct from drug resistance. Drug resistance allows bacteria to grow at drug concentrations that kill their susceptible siblings. In contrast, drug tolerance prolongs survival during exposure to a bactericidal drug concentration without changing the minimal concentration of the drug that is required to prevent growth[4]. In vivo, the most tolerant forms of *Mtb* are located within the avascularized lung caseum[5–8]. Additionally, drug tolerance facilitates the emergence of drug resistance[9]. Targeting *Mtb* drug tolerance is thus not only a promising strategy to shorten TB chemotherapy[10] but may also slow or prevent the emergence and spread of drug-resistant TB.

Isoniazid is an essential component of the current front-line treatment regimen for drug-sensitive TB which also includes rifampicin, pyrazinamide, and ethambutol. Within *Mtb*, isoniazid is converted by the catalase-peroxidase KatG into an isoniazid-NAD adduct[11] representing its major bioactive species[12]. Binding of isoniazid-NAD to the NADH-dependent fatty acid synthase II (FAS-II) component enoyl-acyl carrier protein reductase InhA inhibits mycolic acid biosynthesis[11,13]. In murine models of *Mtb* infection, isoniazid shows effective killing in the acute phase of infection, during which bacteria replicate exponentially inside the lungs. This is followed by a protracted chronic phase during which bacterial numbers stabilize as bacterial replication slows due to onset of adaptive immunity, resulting in a marked loss of activity of isoniazid against *Mtb*[14–16]. In addition, treatment of infected humans and guinea pigs with isoniazid results in a biphasic mode of killing that consists in a rapid initial killing of actively replicating *Mtb* followed by a slower killing of isoniazid-tolerant subpopulations[17,18].

The so-called *c*ompetence *in*ducing gene A (*cinA*), is present in the majority of bacteria and encodes a deamidase that is thought to facilitate the recycling of nicotinamide mononucleotide generated through non-redox utilization of NAD by enzymes such as the DNA ligase LigA[19]. In *Mtb*, as is the case for many bacterial species, this deamidase is fused to an N-terminal COG1058 domain that is endowed with pyrophosphatase activity against adenosine nucleotide-containing molecules such as ADP ribose and NADH[20,21].

Herein, we demonstrate that the pyrophosphatase domain of the *Mtb* CinA protein acts as a modulator of four clinically used TB drugs, namely isoniazid, ethionamide, delamanid, and pretomanid, most likely via a single mechanism, the cleavage of drug-NAD adducts. We report detection of a pretomanid-NAD adduct, which has previously been suggested but not reported to contribute to the mechanism of action of pretomanid[22]. We show that deletion of CinA results in accelerated killing in vitro via reduction of drug-tolerant *Mtb* persisting during treatment despite being intrinsically susceptible to these drugs. In vivo, the absence of CinA potentiates not only isoniazid treatment during the chronic phase of mouse infection, when isoniazid kills less effectively, but also enhances killing by the recently developed pretomanid containing regimen BPaL, which showed promising outcome when used to treat patients with multidrug-resistant *Mtb* in the NIX-TB trial and has been approved by the FDA to treat highly drug-resistant *Mtb* strains[23].

## Results

### Tnseq screens identify *cinA* as a mediator of isoniazid tolerance.

We performed genetic screens for *Mtb* transposon mutants exhibiting increased susceptibility to killing by isoniazid in two conditions in which tolerance to the drug is well documented: adaptation to IFNγ-activated primary mouse bone marrow-derived macrophages (BMDM)[7] and starvation in phosphate-buffered saline (PBS)[24]. In BMDMs, the mutant library was subjected to two cycles of infection. Each time, isoniazid treatment was initiated 24 h post infection at which point the bacteria have been internalized by the macrophages. Comparing the relative abundance of each mutant in the input pool to the mutant pool recovered after the second BMDM infection identified 18 mutants that were significantly underrepresented after intracellular exposure to isoniazid (Supplementary Fig. 1a and Supplementary Data 1). Seventeen of these mutants were also underrepresented following infection of DMSO-treated control BMDMs, leaving a single isoniazid-specific hit with transposon insertions in *cinA* (*rv1901*). Similarly, *cinA* mutants were the only mutants underrepresented with statistical significance after treatment of PBS-starved *Mtb* with isoniazid (Supplementary Fig. 1b).

### CinA confers isoniazid tolerance in macrophages and in vitro.

To confirm that CinA imparts isoniazid tolerance to non-replicating *Mtb*, we constructed a *cinA* deletion mutant (Δ*cinA*) and confirmed loss of CinA by immunoblot (Supplementary Fig. 2a). Both, wild-type *Mtb* and the Δ*cinA* mutant survived equally well during nutrient starvation in PBS and as expected, isoniazid did not kill wild-type *Mtb* in PBS. In contrast, the Δ*cinA* mutant suffered from a marked loss in viability when exposed to isoniazid during PBS starvation (Fig. 1a). The Δ*cinA* mutant was also killed in resting and IFNγ-activated macrophages by a concentration of isoniazid that affected neither wild-type *Mtb* nor the complemented mutant (Fig. 1b, c). In contrast to its impact on isoniazid, deletion of *cinA* did not affect the activity of rifampicin against *Mtb* in PBS or macrophages (Fig. 1a–c).

In standard liquid culture, the minimum inhibitory concentrations (MICs) of isoniazid and rifampicin were similar for the Δ*cinA* mutant and wild-type *Mtb* (Fig. 1d), indicating that CinA does not mediate intrinsic resistance to isoniazid. However, the kinetics of isoniazid-mediated killing in liquid cultures were faster for the Δ*cinA* mutant than for wild-type *Mtb*. Although the kill kinetics maintained the characteristic biphasic pattern revealing phenotypically tolerant persister cells, the persister population was reduced up to 61-fold in the Δ*cinA* mutant compared to wild-type *Mtb* (Fig. 1e).

### Deletion of CinA potentiates isoniazid treatment during chronic mouse infection.

In mice, the Δ*cinA* mutant replicated normally during acute infection and survived equally well as wild-type when *Mtb* entered the chronic phase of infection (Fig. 2a and Supplementary Fig. 3a). In vivo isoniazid treatment leads to effective killing of *Mtb* during the acute phase of infection; however, after chronic infection is established isoniazid becomes substantially less potent[16]. When infected mice were treated with isoniazid for 12 weeks, starting 6 weeks post aerosol infection, the Δ*cinA* mutant was killed more efficiently than wild-type *Mtb* (Fig. 2b–d and Supplementary Fig. 3b–e). Isoniazid at 10 mg/kg/day reduced bacterial burden of wild-type *Mtb* in lungs by 200-fold in 12 weeks, whereas Δ*cinA* mutant titers declined almost 6000-fold from two million CFU at the beginning of isoniazid treatment to 360 CFU (Fig. 2b and Supplementary Fig. 3e). At 5 mg/kg/day viability of the Δ*cinA* mutant decreased by three orders of magnitude (1200-fold) while wild-type *Mtb* decreased

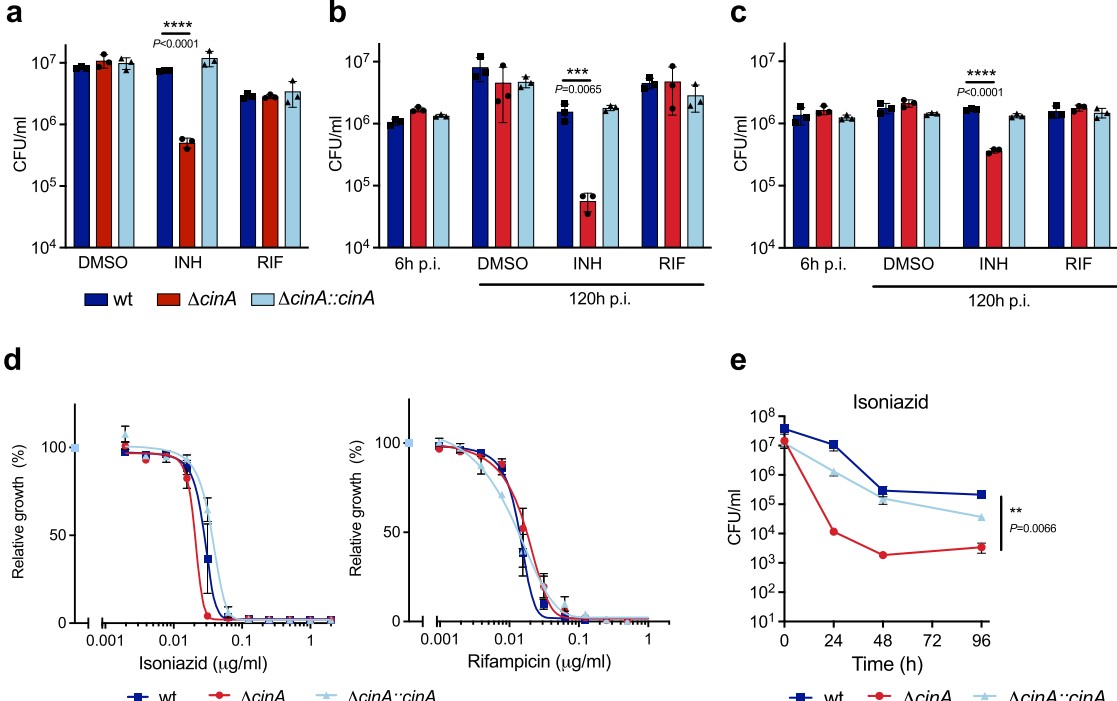

**Fig. 1 CinA mediates intrinsic isoniazid tolerance. a** Bacteria were starved in PBS for 14 days and then exposed to isoniazid (INH, 0.5 μg/ml), rifampicin (RIF, 1 μg/ml), or an equal amount of DMSO for 7 days and cultured for CFU enumeration. Data are means ± SD of triplicate cultures and are representative of two independent experiments. **b** CFU from resting and **c** IFN-γ activated primary murine BMDMs infected with the indicated strains and treated with INH (0.1 μg/ml), RIF (0.1 μg/ml), or an equal amount of DMSO from 24 to 120 h post infection. Data are means ± SD of triplicate cultures and are representative of two independent experiments for **c**. **d** Impact of isoniazid and rifampicin on growth of the indicated strains. Data are means ± SEM from two independent experiments, each performed with duplicate cultures. **e** CFU quantification of the indicated strains after incubation with 0.5 μg/ml isoniazid in standard growth media. Data are means ± SEM from two independent experiments each performed with triplicate cultures. Statistical significance of the differences between wild-type and Δ*cinA* was assessed by two-tailed, unpaired *t*-test, \*\**P* < 0.01, \*\*\**P* < 0.001, \*\*\*\**P* < 0.0001. Source data are provided as a Source Data file.

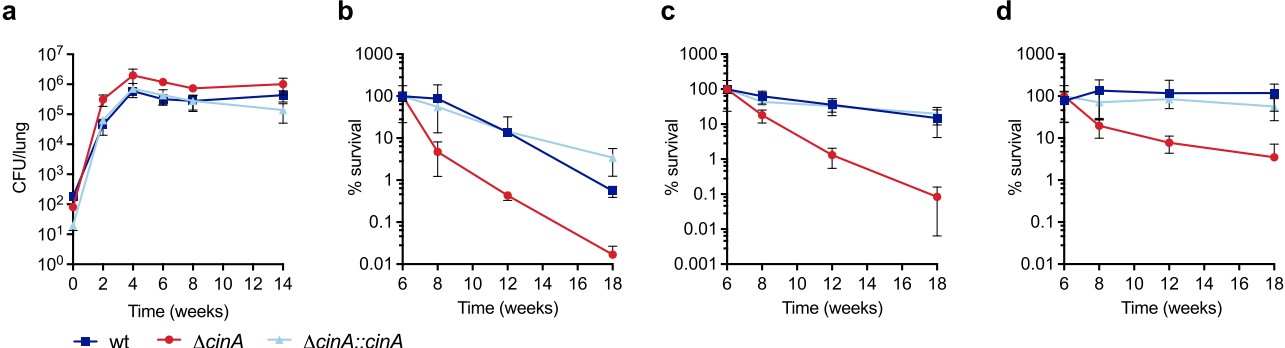

**Fig. 2 Deletion of *cinA* potentiates the efficacy of isoniazid during chronic mouse infection. a** Bacterial titers from lungs of C57BL/6 mice infected with the indicated strains by aerosol. **b–d** Normalized CFU in lungs of C57BL/6 mice infected with the indicated strains and treated with isoniazid at **b** 10 mg/kg/day, **c** 5 mg/kg/day, **d** 2.5 mg/kg/day starting after 6 weeks of infection. CFU are from lung homogenates and normalized to the beginning of isoniazid treatment. Mice received isoniazid in drinking water ad libitum. Data are means ± SD of four mice (except in **a** week 14, Δ*cinA* n = 3 and in **d** week 12, wild-type n = 3) and are representative of two independent infections. In the second experiment drug treatment started after 4 weeks of infection with similar impact on bacterial survival as observed here. Source data are provided as a Source Data file.

less than 10-fold (Fig. 2c and Supplementary Fig. 3e). Even isoniazid concentrations as low as 2.5 mg/kg/day, which did not affect the viability of wild-type *Mtb* at all, markedly reduced viability of the Δ*cinA* mutant (Fig. 2d and Supplementary Fig. 3e). This is relevant because standard doses of isoniazid failed to sustain therapeutic levels in the majority of distinct pulmonary lung lesions including caseum where isoniazid-tolerant persisters reside[25]. The heightened susceptibility to isoniazid was also observed for Δ*cinA* mutant bacilli residing in mouse spleens

(Supplementary Fig. 3b–e) and has been observed in a study that screened transposon mutants for altered susceptibility to the first-line antibiotics in mice[26]. Clearly, deletion of *cinA* potentiates the activity of isoniazid during chronic infection in mice.

**CinA requires an intact pyrophosphatase domain to confer tolerance to isoniazid, ethionamide, delamanid, and pretomanid.** As CinA consists of two independent functional domains, an N-terminal COG1058 domain and a C-terminal

PncC domain, we sought to identify which domain confers isoniazid tolerance. We complemented the $\Delta cinA$ mutant with CinA proteins containing point mutations that abrogate the enzymatic activity of either the COG1058 pyrophosphatase domain ($CinA_{D80A}$) or the PncC deamidase domain ($CinA_{K323A}$) (Supplementary Fig. 2b)[20,21,27]. Both $CinA_{D80A}$ and $CinA_{K323A}$ were well expressed in the $\Delta cinA$ mutant (Supplementary Fig. 2b), but only $CinA_{K323A}$ complemented the tolerance defect of PBS-starved $\Delta cinA$ (Fig. 3a). This suggested that isoniazid tolerance mediated by CinA is due to its pyrophosphatase activity.

The NADH pyrophosphatase (NudC) of *M. bovis* can cleave the isoniazid-NAD adduct into AMP and isoniazid-nicotinamide mononucleotide, and its expression in *M. smegmatis* caused resistance to isoniazid and ethionamide[28]. In *Mtb* NudC is inactive due to a polymorphism at a highly conserved residue, P237, that is critical for activity[28]. We speculated that the pyrophosphatase domain of CinA might confer tolerance to isoniazid through direct cleavage of the isoniazid-NAD adduct. If so, CinA might also mediate tolerance to other antibiotics whose activity depends on the presence of an NAD adduct, such as ethionamide[29]. Indeed, ethionamide killed PBS-starved $\Delta cinA$ bacilli but did not affect viability of PBS-starved wild-type *Mtb* (Fig. 3a). Two other drugs approved for treatment of TB were recently shown (delamanid) or speculated (pretomanid) to act through an NAD adduct[22]. Delamanid and pretomanid both killed non-replicating $\Delta cinA$ bacilli better than wild-type *Mtb*; however, the differences observed for pretomanid were not as drastic as those observed for the other three drugs (Fig. 3a).

*Mtb* residing in caseum is extremely drug tolerant[8]. Therefore, we tested the impact of isoniazid and pretomanid on $\Delta cinA$ residing in a caseum mimetic[30]. Following a 4-week incubation in lipid-rich caseum mimetic, non-replicating bacteria were treated with increasing concentrations of isoniazid and pretomanid for 7 days (Fig. 3b). The minimal bactericidal concentrations of isoniazid and pretomanid required to reduce viability of $\Delta cinA$ 10-fold ($MBC_{90}$) were 10- and 15-fold lower, respectively, compared to the $MBC_{90}$ against wild-type *Mtb* and the complemented strain. Of note, the impact of deletion of *cinA* on the pretomanid bactericidal activity was concentration dependent. Above 8 μM, the difference in kill kinetics between wild-type and $\Delta cinA$ disappeared.

As observed for isoniazid, deletion of *cinA* also accelerated killing of replicating *Mtb* by ethionamide, delamanid and, to a lesser extent, pretomanid, but not rifampicin (Fig. 3c). Because the MIC of pretomanid was approximately fourfold lower against $\Delta cinA$ than wild-type *Mtb* (Fig. 3d), we used pretomanid at two concentrations, 15 × MIC against $\Delta cinA$ (1 μg/ml) and 15 × MIC against wild-type (4.5 μg/ml). The difference in pretomanid kill kinetics was concentration dependent similar to what we observed in the caseum mimetic; at 1 μg/ml pretomanid selectively killed $\Delta cinA$ but did not affect viability of wild-type *Mtb*, while increasing the pretomanid concentration to 4.5 μg/ml resulted in a similar killing of wild-type and $\Delta cinA$ (Supplementary Fig. 4a). Prolonged exposure to pretomanid and isoniazid demonstrated that drug-resistant mutants emerged around the same time (Supplementary Fig. 4a, b). However, since $\Delta cinA$ was killed more rapidly before resistant mutants took over, the absolute number of resistant $\Delta cinA$ bacteria was smaller than the number of resistant wild-type *Mtb* at every time point.

The MICs of rifampicin, isoniazid, ethionamide, and delamanid in liquid culture were not significantly affected by deletion of *cinA* (Fig. 3d and Supplementary Table 1). Increased killing of $\Delta cinA$ by these drugs was thus due to a reduction in drug tolerance. In the case of pretomanid a small MIC shift (~4.5-fold) was observed indicating increased susceptibility of the mutant which might be a result of CinA contributing to intrinsic

resistance to that drug (Fig. 3d and Supplementary Table 1). All phenotypes of $\Delta cinA$ were complemented by CinA and $CinA_{K323A}$ but not $CinA_{D80A}$ (Fig. 3 and Supplementary Table 1). Thus, CinA, via its pyrophosphatase domain, confers tolerance to isoniazid, ethionamide, delamanid, and partial resistance to pretomanid.

**CinA overexpression increases isoniazid tolerance**. To assess the impact of CinA overexpression on Mtb drug tolerance and resistance, we expressed *cinA* under the control of an anhydrotetracycline (atc) inducible promoter (*cinA*-TetON). This resulted in CinA overexpression in both the absence and presence of atc, likely because of leaky transcription by the TetR-controlled promoter (Fig. 4a). *CinA*-TetON was as tolerant to isoniazid as wild-type *Mtb* and in the presence of atc *cinA*-TetON survived isoniazid exposure without loss of viability while CFU of wild-type and the uninduced *cinA*-TetON declined 10-fold (Fig. 4b). Treatment of *cinA*-TetON with atc resulted in a significant growth defect (Supplementary Fig. 4c, d) and did not increase the MIC of isoniazid more than observed with the uninduced strain. We therefore determined MICs of other drugs using *cinA*-TetON without the addition of atc. This demonstrated that CinA overexpression resulted in a small and reproducible increase of the isoniazid and ethionamide MICs, but did not affect the MICs of delamanid, pretomanid and rifampicin compared to those against wild-type *Mtb* (Fig. 4c).

**Deletion of CinA results in accumulation of NAD adducts in isoniazid and ethionamide treated *Mtb***. To substantiate the hypothesis that CinA is capable of cleaving NAD-drug adducts, we measured the impact of deleting *cinA* on the accumulation of isoniazid-NADH adducts in *Mtb*. We first exposed *Mtb* to isoniazid and prepared extracts for analysis by liquid chromatography-coupled tandem mass spectrometry (LC-MS/MS). This analysis revealed a molecule that: (i) matched both the predicted mass of the isoniazid-NAD adduct ($[M + H] = 771.153$) and observed chromatographic retention time and mass of a chemically synthesized isoniazid-NADH standard with a mass error <10 ppm, (ii) was only observed in isoniazid-treated samples; and (iii) exhibited a fragmentation pattern consistent with its predicted chemical structure and observed with a chemical standard synthesized by chemical oxidation of INH with stoichiometric amounts of manganese pyrophosphate and NAD(H) (Fig. 5a)[31,32]. The observed high-resolution fragmentation pattern specifically included mass ions corresponding to the adenosine diphosphate ($[M + H] = 428.03540$) and adenine ($[M + H] = 136.063$) moieties of the isoniazid-NAD adduct. The intensity of these peaks increased in a dose-dependent fashion, with the mutant accumulating approximately threefold higher levels of adduct than wild-type (Fig. 5b). As anticipated, this accumulation was only restored to wild-type levels by complementation with a CinA allele harboring a functional pyrophosphatase domain (Fig. 5c) and overexpression of CinA resulted in a further reduction in isoniazid-NAD adduct levels (Supplementary Fig. 4e). *Mtb* exposed to ethionamide revealed a similar mass corresponding to an ethionamide-NAD adduct—fragmentation of which also confirmed the presence of an adenosine diphosphate and adenine moiety—suggesting the mutant also accumulated higher levels of the ethionamide-NAD adduct than wild-type *Mtb*[28] (Supplementary Fig. 5).

Given the apparent role of CinA-mediated cleavage of NAD-drug adducts in tolerance to both isoniazid and ethionamide, we tested the role of this activity in mediating tolerance to the nitroimidazole class of anti-tubercular compounds. To do so, we exposed *Mtb* to pretomanid and sought evidence of molecules that met the following criteria: they were specific to pretomanid-

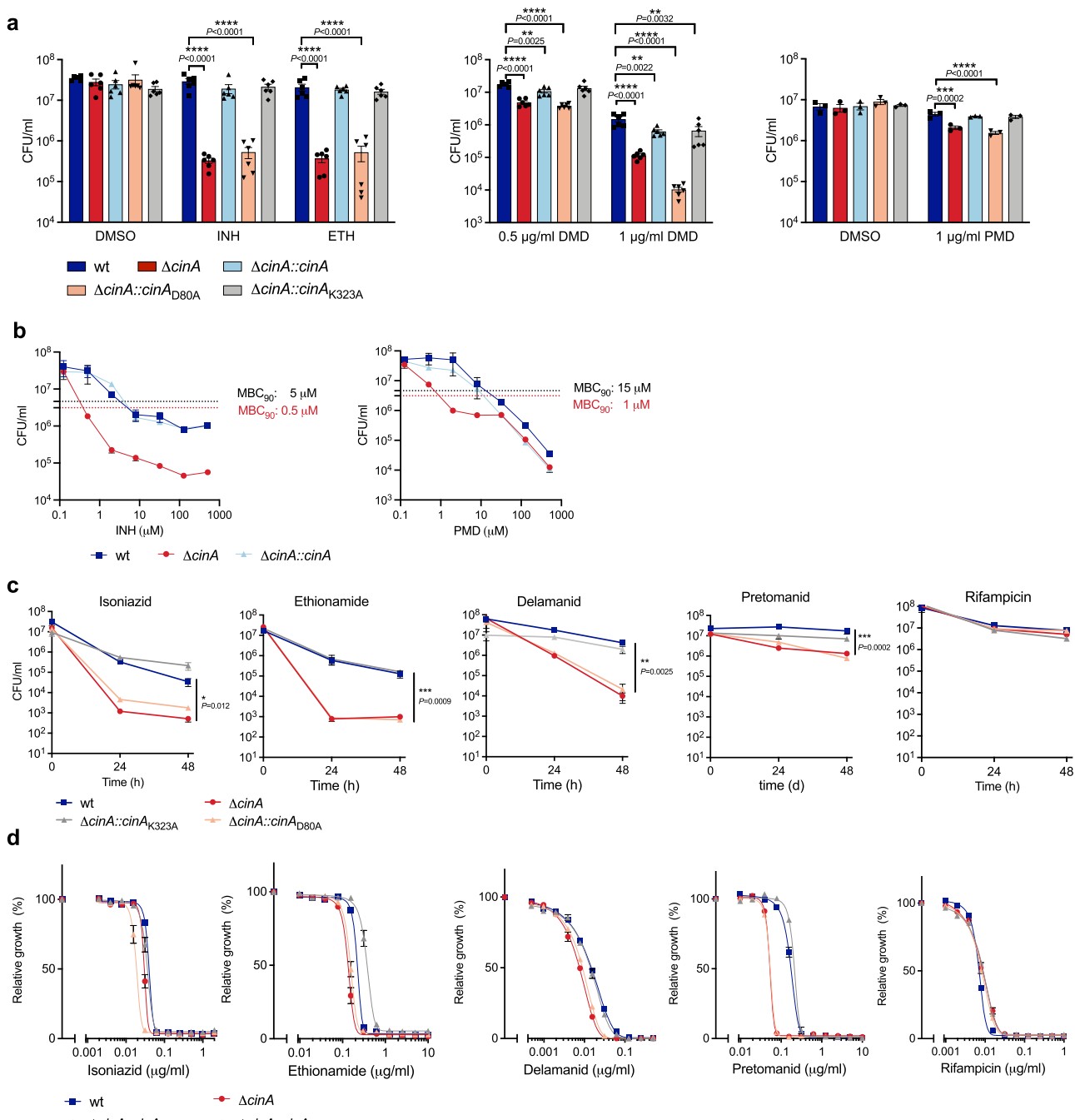

**Fig. 3 CinA requires an intact pyrophosphatase domain to confer tolerance to isoniazid, ethionamide, delamanid, and pretomanid. a** Bacteria were starved in PBS for 21 days and then exposed to 0.5 µg/ml isoniazid (INH), 50 µg/ml ethionamide (ETH), or were starved for 14 days and then exposed to 0.5 µg/ml and 1 µg/ml delamanid (DMD), 1 µg/ml pretomanid (PMD) or equal amount of DMSO for 7 days and cultured on agar plates for CFU enumeration. Data are means ± SEM from two independent experiments each with triplicate cultures, except for PMD data, which are from one experiment with triplicate cultures. Statistical significance was assessed by one-way ANOVA followed by Dunnett's multiple comparison test. **$P < 0.01$, ***$P < 0.001$, ****$P < 0.0001$. **b** CFU quantification of the indicated strains after incubation in caseum mimetic for 4 weeks followed by 1 week treatment with isoniazid or pretomanid at the indicated concentrations. The stippled lines indicate a 10-fold reduction of wild-type and $\Delta cinA$ relative to DMSO-treated cultures and were used to determine the $MBC_{90}$ for these strains. Data are means ± SD of triplicate cultures and representative of two experiments. **c** CFU quantification of the indicated strains after incubation with 0.5 µg/ml isoniazid, 4.5 µg/ml ethionamide, 50 µg/ml delamanid, 1 µg/ml pretomanid, and 5 µg/ml rifampicin in standard growth media. Data are means ± SEM from one or two independent experiments each with triplicate cultures. Statistical significance of the difference between wild-type and $\Delta cinA$ was assessed by two-tailed, unpaired $t$-test, *$P < 0.05$; ****$P < 0.0001$. **d** Impact of antibiotics on growth of the indicated strains. Data are means ± SEM of triplicate cultures and representative of at least two experiments. Source data are provided as a Source Data file.

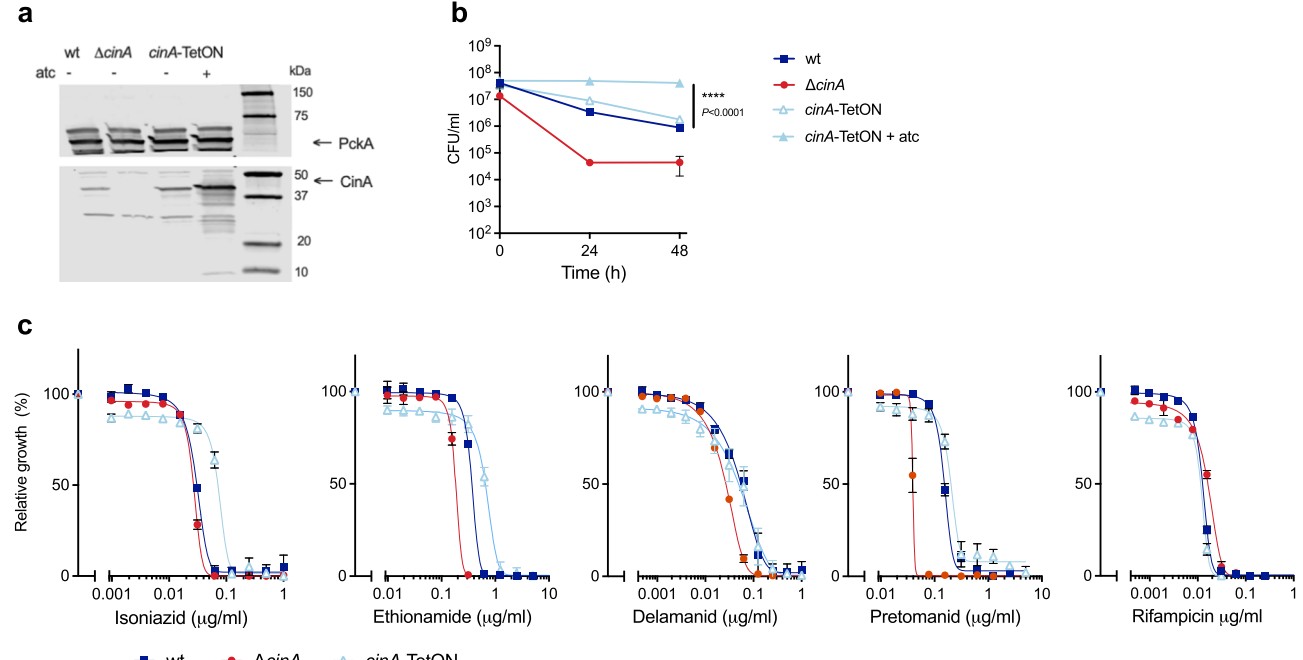

**Fig. 4 CinA overexpression increases isoniazid tolerance. a** Immunoblot analysis of CinA in wild-type, Δ*cinA*, and *cinA*-TetON strains. PckA serves as a loading control. The experiment was performed three times with similar results. **b** CFU quantification of the indicated strains after incubation with 0.5 μg/ml isoniazid in standard growth media. Data are means ± SD from triplicate cultures and representative of two experiments. Statistical significance of the difference between wild-type and *cinA*-TetON +atc was assessed by two-tailed, unpaired *t*-test, ****$P < 0.0001$. **c** Impact of antibiotics on growth of the indicated strains. Data are means ± SD of triplicate cultures and representative of at least two experiments. Source data are provided as a Source Data file.

treated *Mtb*, as reported by their absence in both untreated cells and cells treated with INH or ETH; displayed a dose-dependent increase in cells exposed to pretomanid and were structurally consistent with adduction to a NAD molecule. We identified one such ion ([M + H] = 979.1995) that matched these criteria and yielded fragmentation products corresponding to an in silico prediction of a putative pretomanid-NAD adduct[33], as a chemical standard could not be synthesized (Fig. 5d, e). Analogous to the isoniazid- and ethionamide-NAD adducts, this included fragmentation products corresponding to the adenine, adenosine, and nicotinamide riboside-pretomanid moieties of NAD/H ([M + H] = 136.062, 250.093, 632.124, respectively)[33]. Relative quantitation of the pretomanid-NAD adduct in pretomanid-treated *Mtb* revealed accumulation in Δ*cinA*, which was reduced in the complemented strain. *Mtb* expressing CinA_{D80A} (non-functional pyrophosphatase domain) accumulated pretomanid-NAD similar to Δ*cinA*, while expression of CinA_{K323A} (with an inactive PncC deamidase domain) resulted in levels similar to those in the complemented strain (Fig. 5e). These results were consistent with those obtained in isoniazid-treated *Mtb* (Fig. 5c) and corresponded to the phenotypic susceptibility of these strains to pretomanid (Fig. 3c, d). Taken together, these results support a model in which CinA mediates tolerance to isoniazid, ethionamide, and the nitroimidazole pretomanid via cleavage of their corresponding NAD adducts through its pyrophosphatase domain.

**Absence of CinA potentiates the efficacy of BPaL.** The Nix-TB trial revealed that short course treatment with a combination of bedaquiline, pretomanid, and linezolid (BPaL) had a favorable outcome in 90% of patients with multidrug-resistant TB[23]. To test if CinA inactivation affects the efficacy of BPaL, we infected mice with wild-type *Mtb* or Δ*cinA* and 4 weeks later treated them with a BPaL regimen for 4 weeks (Fig. 6 and Supplementary Fig. 6a). This resulted in substantially more killing of Δ*cinA* relative to

wild-type *Mtb* especially in mouse lungs where Δ*cinA* CFU declined over 100-fold, while wild-type *Mtb* CFU declined 10-fold. CinA deletion thus also potentiates the efficacy of BPaL, an FDA-approved regimen for multidrug-resistant forms of TB, in mice.

## Discussion

Isoniazid, a corner stone of standard *Mtb* drug treatment, kills replicating bacilli in vitro and in vivo. However, when *Mtb* enters a slow- or non-replicating state, for example, due to nutrient starvation, hypoxia, or in response to adaptive immunity, the entire population becomes isoniazid tolerant and persists even in the presence of high isoniazid concentrations[24,34]. Little is known about the mechanisms leading to isoniazid tolerance when *Mtb* ceases to replicate. To identify genetic factors that aid the survival of *Mtb* in conditions that render the whole population refractory to isoniazid treatment we performed genome-wide screens and identified a single gene, *cinA*.

Deletion of *cinA* or inactivation of CinA's pyrophosphatase domain not only increased killing in conditions where the entire *Mtb* population exhibits isoniazid tolerance, it also resulted in a substantially smaller persister subpopulation within actively replicating *Mtb* cultures. During axenic growth isoniazid rapidly kills more than 99% of the replicating *Mtb* population; however, a small subpopulation of isoniazid persister cells is only killed slowly resulting in a biphasic kill curve[35]. Stochastic repression of KatG activity, asymmetric cell division, and reversible increases in drug efflux or posttranslational modification can all lead to isoniazid-tolerant persisters[36–40]. The fraction of persisters present in replicating *Mtb* may thus consist of yet smaller, mechanistically distinct subpopulations. These subpopulations refractory to drug treatment are of particular interest as they are thought to contribute to the unusually long TB treatment and targeting these subpopulations has the potential to shorten TB chemotherapy[10]. The absence of CinA did not prevent the

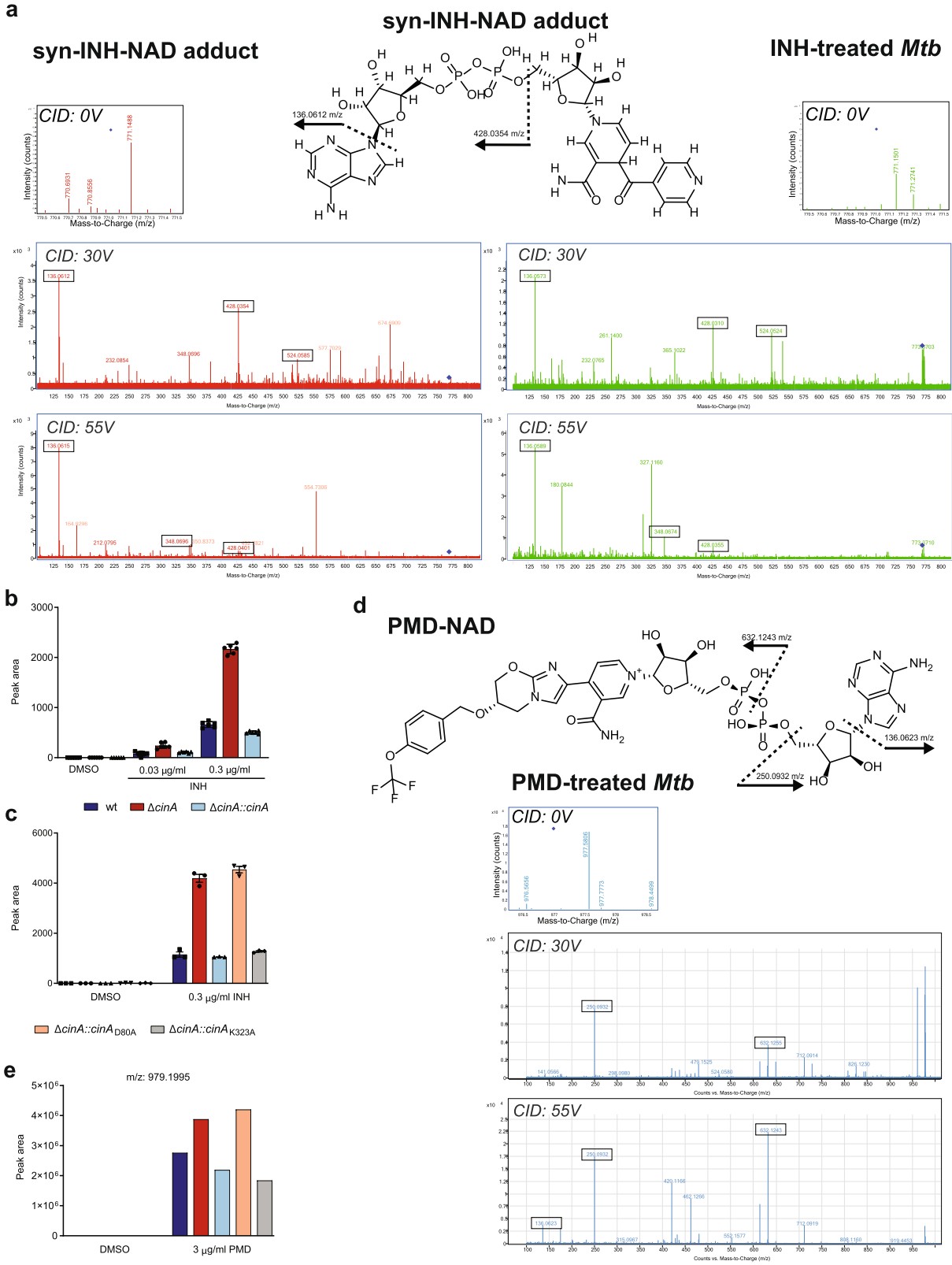

emergence of drug-resistant bacteria, but the drug-resistant population of Δ*cinA* was smaller than the one that emerged in drug-treated wild-type *Mtb*. This suggests that CinA inactivation could delay the appearance of drug-resistant *Mtb*. This work also underscores that drug tolerance does not always rely on slowly or non-growing cells, because the pyrophosphatase activity of CinA impaired drug mediated killing of actively replicating *Mtb*.

CinA mediated tolerance not only to isoniazid but also to ethionamide, delamanid, and pretomanid. The MICs of isoniazid, ethionamide, and delamanid were not different between wild-type *Mtb* and the *cinA* mutant, indicating that CinA mediates tolerance and not resistance to these drugs. In the case of pretomanid, however, a small MIC shift was observed leaving the possibility that CinA also confers intrinsic resistance to this drug. Moreover,

**Fig. 5 Inactivation of CinA results in accumulation of NAD adducts in isoniazid (INH) and pretomanid (PMD) treated *Mtb*. a** MS/MS fragmentation spectra of a chemically synthesized isoniazid-NAD adduct and corresponding metabolite observed only in isoniazid-treated *Mtb* with the predicted mass of the INH-NAD adduct. Structures and annotated fragments provide confirmatory evidence of mass matching to the predicted parent adduct (Δppm = 10) and fragments corresponding to the ADP and adenine moieties of the adduct. Boxed masses additionally indicate matching collision energy-dependent fragmentation products observed in both the chemical standard and INH-treated *Mtb*. Additional non-matching masses correspond to likely fragmentation products of co-gating parent masses that entered the collision cell with the INH-NAD adduct as observed in the 0 V spectra. **b** Quantification of the isoniazid-NAD adduct as a function of the isoniazid concentration that the indicated strains were exposed to. **c** Accumulation of isoniazid-NAD is reversed by expression of CinA with a functional pyrophosphatase domain. Data are means ± SD of six (**b**) or three (**c**) cultures. **d** MS/MS fragmentation spectra of a mass ion corresponding to the predicted mass of a PMD-NAD adduct observed in pretomanid-treated *Mtb*. Structures and annotated masses provide confirmatory evidence of mass matching to the predicted parent adduct (Δppm = 10) and fragments corresponding to the adenine, adenosine, and nicotinamide riboside-pretomanid moieties of the PMD-NAD adduct. **e** Quantification of the relative abundance of pretomanid-NAD adduct in the indicated strains. Data are from six independent cultures that were pooled to ensure robust detection of potential differences in adduct levels. Source data are provided as a Source Data file.

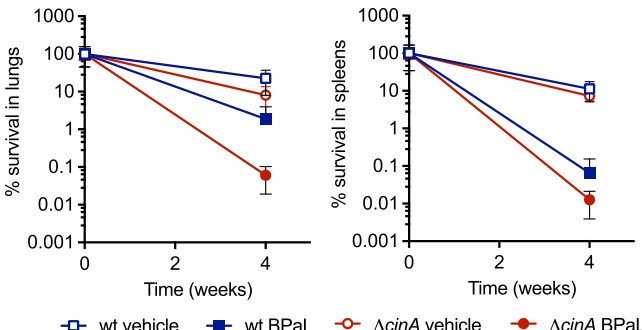

**Fig. 6 Deletion of *cinA* potentiates the efficacy of BPaL.** Balb/C mice were infected with wild-type and Δ*cinA* and treated with BPaL (bedaquiline 20 mg/kg/day, pretomanid 50 mg/kg/day, and linezolid 100 mg/kg/day) starting after 4 weeks of infection. CFU recovered from lungs and spleens are normalized to the beginning of BPaL treatment. Data are means ± SD of six mice. Source data are provided as a Source Data file.

overexpression of CinA increased the MICs of isoniazid and ethionamide but had no impact on the activities of pretomanid or delamanid. All four drugs are pro-drugs whose activation requires a bacterial enzyme. For isoniazid, ethionamide, and delamanid it has been demonstrated that activation leads to NAD-drug adducts; adduct formation of pretomanid has been predicted because of the similarity of this drug to delamanid[11,22,29]. We confirmed this prediction by identifying the pretomanid-NAD adduct in live *Mtb*. This supports our hypothesis that CinA mediates tolerance through cleavage of formed NAD-drug adducts, which is further substantiated by our finding that in the absence of CinA NAD-drug adducts accumulated inside the bacilli during isoniazid, ethionamide, and pretomanid treatment. The different impact of CinA expression on drug tolerance and resistance suggest that the antibiotics we tested inhibit growth and kill *Mtb* by more than one mechanism and that the NAD adduct is more important for killing than growth inhibition. Similarly, the concentration-dependent hypersusceptibility of Δ*cinA* to pretomanid implies that the pretomanid-NAD adduct contributes to its bactericidal activity at low concentrations, while at higher concentrations other mechanisms dominate.

In this study we identified a single protein, CinA, as a mediator of tolerance to four TB drugs in clinical use. CinA-mediated tolerance is a direct result of its pyrophosphatase domain facilitating cleavage of NAD-drug adducts. Hence, targeting CinA with inhibitors of its pyrophosphatase domain has the potential to shorten treatment of drug-sensitive TB by potentiating the first-line drug isoniazid or enhancing treatment regimens for multi- and extensively drug-resistant TB by potentiating delamanid and pretomamid. Furthermore, targeting CinA could also

complement and extend ongoing efforts to improve the clinical value of ethionamide containing regimens[41,42]. Clinical precedents for this strategy exist, for example, in the form of Zavicefta in which the β-lactamase inhibitor avibactam strongly potentiates the activity of the cephalosporin ceftazidime[43] to the extent that it is effective against *Mtb* including highly drug-resistant strains[44].

## Methods

**Bacterial strains and media**. *Mycobacterium tuberculosis* H37Rv was used as the parental strain for all mutants throughout the study. Bacteria were cultured using Middlebrook 7H9 liquid medium supplemented with 0.2% glycerol, 0.05% Tween-80, and ADN (0.5% bovine serum albumin, 0.2% dextrose, 0.085% NaCl) or Middlebrook 7H10 solid agar supplemented with 0.2% glycerol, 10% Middlebrook oleic acid-albumin-dextrose-catalase (OADC) (Becton, Dickinson) and if necessary supplemented with 0.4% activated charcoal. Antibiotics for selection of genetically modified strains were added at the following concentrations: hygromycin (50 μg/ml) and kanamycin (25 μg/ml).

**Drugs**. Isoniazid (INH), Rifampicin (RIF), Ethionamide (ETH), and Pretomanid (PMD) were purchased from Sigma; Delamanid (DMD) was procured from MedKoo Biosciences. All antibiotics were dissolved in DMSO with exception of INH, which when used for treatment of mice, was dissolved in $H_2O$.

**Preparation of murine BMDMs**. Bone marrow cells were harvested from female C57BL/6 mice and differentiated into BMDMs by culturing the cells in the presence of 20% L-cell conditioned medium[45]. Macrophages were seeded at $1 \times 10^7$ cells per T75 flask or $1 \times 10^5$ per well in 96-well plates in Dulbecco's modified Eagle's medium supplemented with 10% fetal bovine serum, 10% L929 culture filtrate, and 10 mM HEPES overnight prior to infection. Where indicated macrophages were activated with IFNγ (20 ng/ml) overnight prior to infection and maintained in IFNγ for the remainder of the infection. Bacteria were grown to early log phase, washed in PBS-Tween-80 0.05% prior to being diluted into DMEM, and added to macrophages at a multiplicity of infection (MOI) of 0.1. After 4 h macrophages were washed twice with warm PBS to remove extracellular bacteria. Intracellular bacteria were enumerated by lysing macrophages with 0.01% Triton X-100 and plating serial dilutions of lysates on 7H10 agar supplemented with 0.4% charcoal.

**TnSeq screen**. Transposon mutant libraries were constructed in wild-type *Mtb* H37Rv by himar1 mutagenesis[46,47]. Transposon mutant library stocks were cultured in liquid medium for 4 days with agitation (100 rpm) to allow recovery. For PBS starvation experiments cultures were washed twice in PBS containing 0.05% tyloxapol and adjusted to an optical density (OD)$_{580}$ of 0.2. Libraries were starved for 14 days prior to exposure to 0.5 μg/ml INH or an equal amount of DMSO. After a total of 28 days of starvation libraries were cultured on 7H10 agar supplemented with 0.4% charcoal.

For infection of BMDMs $1 \times 10^7$ IFNγ-activated BMDMs were infected with transposon mutant libraries at a MOI of 1. At 24 h post infection (p.i.) 0.1 μg/ml INH or an equivalent amount of DMSO was added to infected BMDMs. Macrophages were lysed 120 h p.i. and lysates cultured on 7H10 agar supplemented with charcoal. After 21 days of outgrowth colonies recovered from infected macrophages were collected, transferred into liquid medium and grown for 4 days prior to re-infection of BMDMs as before.

Genomic DNA was extracted from the recovered transposon mutant libraries and transposon–chromosome junctions were amplified and sequenced to determine the relative abundance of each transposon mutant in input and INH-treated libraries as described previously[46,48]. Sequencing reads were processed and

analyzed using TPP and TRANSIT tools from the TRANSIT TnSeq analysis platform, respectively, and transposon–chromosome junctions were mapped to the *Mtb* H37Rv reference genome (GenBank accession number NC_018143.1)[49].

**Mutant generation and complementation**. The ΔcinA mutant was constructed in *Mtb* by replacing *cinA* (*rv1901*) with a hygromycin cassette through homologous recombination as described elsewhere[50,51].The first 100 bp of the open reading frame of *cinA* were retained to avoid disruption of the *rv1900c* promoter. Specifically, nucleotides 101–1193 of *cinA* were replaced with a hygromycin-resistance cassette. To achieve this, *Mtb* H37Rv transformed with pNit-RecET-sacB-kanR was grown to mid log phase and expression of the RecET proteins was induced by addition of 10 µM isovaleronitrile. After 8 h of RecET induction the cells were treated with glycine and incubated for 16 h as described[51]. A DNA fragment containing the hygromycin-resistance cassette flanked by 500 bp of the upstream region including the first 100 bp of *cinA* and 500 bp of the sequence downstream of *cinA* was synthesized (GeneScript Biotech) and transformed by electroporation. Knockout candidates were selected on hygromycin-containing agar. Deletion of *cinA* was confirmed by PCR and western blot analysis. Plasmid pNitET-sacB-kan was cured by counterselection on 10% sucrose containing agar. A complemented strain was obtained by introducing a copy of wild-type *cinA* into the *Mtb* genome at the attL5 attachment site under the control of a hsp60 promoter. For this, we constructed pGMCK-P$_{hsp60}$-*cinA*-containing a kanamycin resistance gene, the hsp60 promoter upstream of *cinA* (nucleotides −28 to 1193), the integrase gene, and the phage attP sites for site-specific integration into the mycobacterial chromosome. We used plasmids pDE43-MCK (kanR, multisite Gateway destination plasmid), pEN41A-T02 (ampR multisite Gateway entry plasmid), pEN12A-Phsp60 (ampR, multisite Gateway entry plasmid; containing a hsp60 promoter)[52], and pEN23A-cinA (ampR, multisite Gateway entry plasmid; containing *cinA*, nucleotides −28 to 1193). Complementation strains expressing *cinA* harboring point mutations (D80A = t239g; K323A = aag → gcg 967) rendering either the pyrophosphatase or the deamidase domain of the gene inactive were obtained by the same approach[20,27]. For overexpression we cloned *cinA* downstream of the anhydrotetracycline inducible promoter P606 (ref. [53]). All DNA constructs were obtained from GenScript Biotech and plasmids were generated using Gateway Cloning Technology (Life Technologies).

**Mouse infection**. The mouse experiments to assess virulence of the ΔcinA mutant and the impact of INH during infection were conducted following guidelines for care and use of laboratory animals provided by the National Institute of Health and with approval from the Institutional Animal Care and Use Committee of Weill Cornell Medicine (IACUC protocol 060-441A). Female 8-week-old C57BL/6 mice (Jackson Laboratory) were infected with ~100–200 CFU/mouse using an Inhalation Exposure System (Glas-Col). The mice were housed in HEPA-filtered cages within the ABSL-3 facility with a 12 h light–dark cycle at 70F ± 2F with 50% humidity (range is 30–70%). At indicated time points bacterial burden in lungs and spleens was determined by culturing serial dilutions of organ homogenates on 7H10 agar. Where indicated, mice were treated with isoniazid delivered through drinking water at the following concentrations: 10, 5, and 2.5 mg/kg/day starting from day 42 p.i. Isoniazid-containing water was replaced every 7 days.

The comparative efficacy study of bedaquiline (BDQ)–pretomanid (PMD)–linezolid (LZD) in mice infected with wild-type and ΔcinA H37Rv was conducted following guidelines for care and use of laboratory animals provided by the National Institute of Health and with approval from the Institutional Animal Care and Use Committee of Hackensack Meridian Health (IACUC protocol 265). Female 8-week-old BALB/c mice (Charles River Laboratories) were infected as described above. At 4 weeks p.i., mice received daily oral doses (7 days a week) of BDQ 20 mg/kg, PMD 50 mg/kg, and LZD 100 mg/kg, formulated in 20% HPBCD in 50 mM sodium citrate pH 3, 10% HPBCD/10% lecithin/80% water and 0.5% CMC/0.5% Tween-80, respectively. BDQ and PMD were given in the morning and LZD 4 h later as described in Xu et al.[54]. After 4 weeks of treatment, bacterial burden in lungs and spleens was determined by culturing serial dilutions of organ homogenates on 7H10 agar. After 1 and 3 weeks of daily drug administration, therapeutic drug monitoring was performed by sampling blood from the tail vein of three to six mice per group, at peak and trough, as indicated. Therapeutic drug monitoring demonstrated consistent drug levels across animals within a group, and across mice infected with wild-type *Mtb* and ΔcinA excluding pharmacokinetic variability as a potential confounding factor in interpreting the efficacy data (Supplementary Fig. 5b).

**Drug susceptibility profiling and time-kill assays**. Bacteria were grown with agitation (100 rpm) to mid log phase and washed once prior to preparation of single-cell suspensions in 7H9 liquid medium supplemented with 0.2% glycerol, 0.05% Tween-80, and ADN (0.5% bovine serum albumin, 0.2% dextrose, 0.085% NaCl). For drug susceptibility profiling cell suspensions were adjusted to an OD$_{580}$ of 0.02 and 50 µl culture was added to 384-well plates containing drugs and incubated for 7–12 days. OD$_{580}$ values were determined using an M2 SpectraMax Microplate reader and final values were normalized to no-drug controls. Compounds were dispensed using a D300e Digital Dispenser (HP) with all wells normalized to a final DMSO concentrations of 1%. For time-kill assays cell

suspensions were adjusted to an OD$_{580}$ of 0.1 and drugs were added at the following concentrations: INH 0.5 µg/ml, RIF 0.5 µg/ml, ETH 4.5 µg/ml, DMD 50 µg/ml, PMD 1 µg/ml. At indicated time points bacterial survival was determined by culturing serial dilutions on 7H10 agar (cultures were washed twice with PBS, 0.05% Tween-80) or 7H10 agar supplemented with 0.4% charcoal.

**PBS starvation**. Bacteria were grown with agitation (100 rpm) to mid log phase and washed twice in PBS-Tyloxapol 0.05%. Single-cell suspensions were prepared in PBS-Tyloxapol 0.05% and adjusted to an OD$_{580}$ of 0.05. Cells were starved for 14 or 21 days and then exposed to the following drug concentrations INH 0.5 µg/ml, RIF 0.5 µg/ml, ETH 50 µg/ml, PMD 1 µg/ml, DMD 0.5 µg/ml, and 1 µg/ml or equal volume of DMSO. Colony-forming units were enumerated 7 days later by culturing serial dilutions on 7H10 agar supplemented with 0.4% charcoal.

**Exposure to caseum mimetic and drug treatment**. Bacteria were grown to mid log phase and inoculated into caseum mimetic at 10$^8$ CFU/ml *Mtb* H37Rv (wild-type, ΔcinA and complemented strains), and incubated at 37 °C for 4 weeks in 96-well plates, to allow for physiologic and metabolic adaptation to the lipid-rich environment. The caseum mimetic[30,55] was prepared with the following modifications: to induce lipid droplet formation, PMA-differentiated THP-1 macrophages were infected with irradiated *Mtb* (BEI Resources) at an approximate MOI of 1:50. The foamy macrophages were washed three times with PBS, followed by three freeze–thaw cycles to lyse the cells and incubation at 75 °C for 20–30 min to denature proteins in the matrix[55]. The lysate was allowed to rest at 37 °C for 3 days, prior to freezing at −80 °C until inoculation. After 4 weeks of *Mtb* incubation in the caseum mimetic, drugs were added at concentrations ranging from 0.125 to 512 µM, in fourfold increment, the samples as well as no-drug controls were incubated for 7 days, and serial dilutions were cultured on 7H11 agar medium for 6 weeks prior to CFU enumeration.

**CinA purification and antibody generation**. Full-length *cinA* was cloned into pET28b vector and transformed into BL21(DE) *E. coli*. Bacteria were grown to an OD$_{600}$ of 0.6 in LB media supplemented with 50 µg/ml kanamycin. Subsequently, cultures were cooled to 16 °C and *cinA* expression was induced by addition of 0.5 mM IPTG for 16 h. *E. coli* cells were harvested by centrifugation and lysed by French Press in 20 mM TRIS-HCl pH 7.5, 0.2 M NaCl in the presence of protease inhibitor cocktail (Roche) and DNAase. Lysates containing CinA inclusion bodies were pelleted at 55,000g, washed twice in 20 mM TRIS-HCl pH 7.5, 0.2 M NaCl, 0.1% Tween-80, and dissolved in 20 mM TRIS-HCl pH 7.5 containing 8 M urea. The insoluble fraction of the sample was removed by centrifugation and unfolded protein was purified using a Ni affinity column under denaturing conditions, maintaining 8 M urea throughout all steps. CinA was eluted with 500 mM imidazole. The concentration of the eluted CinA was adjusted to 0.5 mg/ml in 20 mM TRIS-HCl pH 7.5 containing 8 M urea.

The eluate was dialyzed against 50 mM Nacitrate-phosphate pH 7.6, 500 mM NaCl, 1 mM MgCl$_2$, 10% glycerol, 4 M urea overnight at 4 °C. The urea concentration was reduced by step dialysis at room temperature, by twofold diluting the dialysis buffer every 2 h in urea-free buffer until a urea concentration of approximately 0.25 M was reached. Subsequently, the eluate was dialyzed against urea-free buffer overnight at 4 °C. Any precipitated protein was pelleted by centrifugation and the soluble fraction was concentrated to 0.5 mg/ml. Rabbit polyclonal antiserum against recombinant CinA was generated by Thermo Fisher Scientific.

**Affinity purification of polyclonal anti-CinA serum**. To purify CinA-specific antibody from anti-CinA serum 50 µg of purified CinA was resolved through SDS-page and transferred to nitrocellulose membrane. The membrane was incubated with anti-CinA serum (1:6 dilution) at 4 °C overnight. CinA-specific antibody was eluted with 10 mM glycine pH 2.7 for 2 min and the eluate was immediately neutralized with 300 µl of 1.5 M Tris-HCl pH 8.8. Subsequently, the antibody was dialyzed against PBS at 4 °C.

**Immunoblot analysis of CinA**. Protein extracts were prepared from wild-type, ΔcinA, ΔcinA:: cinA, ΔcinA:: cinA$_{D80A}$, ΔcinA:: cinA$_{K323A}$, cinA-TetON strains by mechanical lysis with 0.1 mm zirconia/silica beads in PBS in presence of protease inhibitor cocktail (Roche). Subsequently, lysates were filtered through 0.22 µm Spin-X columns (Corning). Protein concentrations were determined using a DC Protein Assay Kit (Bio-Rad). For immunoblot analysis of CinA, 20 or 40 µg of protein was resolved through SDS-page, transferred to nitrocellulose membrane and probed with anti-CinA serum (1:50 dilution) and anti-PrcB (1:10,000 dilution) or anti-PckA (1:10,000 dilution). Following incubation with anti-rabbit secondary antibody (LI-COR Biosciences), protein bands were visualized using the Odyssey Infrared Imaging System (LI-COR Biosciences).

**Drug metabolite extract preparation**. Drug metabolite extracts of drug-treated *Mtb* were prepared as described[56] with some modifications. Briefly, bacteria were grown to an OD$_{580}$ of 1.0 and 1 ml of culture was inoculated onto 0.22 µm polyvinylidene fluoride membrane filters (Millipore Sigma) using vacuum filtration.

Filters were transferred to 7H10 agar and incubated for 5 days to allow for sufficient biomass to accumulate for drug adduct detection. Subsequently, filters were placed on top of inverted 15 ml centrifuge caps containing 7H9 liquid medium for 24 h. After 24 h the medium inside the caps was replaced with 7H9-containing antibiotics at the following concentrations: INH 0.03 μg/ml or 0.3 μg/ml, ETH 3 μg/ml, PMD 3 μg/ml or an equal volume of DMSO. Small molecules were extracted after 24 h of drug exposure by bead-beating in acetonitrile:methanol:water (40:40:20; v-v:v)[57]. Samples were stored at −80 °C until drug adduct detection by LC-MS/MS.

**Isoniazid-NAD adduct synthesis**. INH-NAD adducts were synthesized by reacting INH and NAD+ in the presence of MnIII pyrophosphate, as described by Nguyen et al.[32]. MnIII pyrophosphate was prepared by stirring an aqueous solution of sodium pyrophosphate decahydrate (200 mM), MnIII acetate dihydrate (50 mM), and pyrophosphoric acid (added to pH 4.5) for 24 h at room temperature[58,59]. Next, NAD+ (0.1 ml of a 20 mM stock; Roche) and INH (0.1 ml of a 20 mM stock) were combined in 0.9 ml of phosphate buffer (100 mM, pH 7.5), followed by the addition of MnIII pyrophosphate (10 consecutive additions of 8 μl of the above solution, 2 min apart, while mixing at 1000 rpm at room temperature). Twenty minutes after the last addition, the reaction mixture was clarified by centrifugation and stored at −80 °C.

**Drug adduct detection**. Putative NAD adducts of isoniazid and ethionamide were separated based on a previously published chromatography method[22,58], using an Agilent 1290 Infinity LC system coupled to a 6545 quadrupole time-of-flight mass spectrometer for high-resolution MS/MS fragmentation analysis and a 6495 triple quadrupole mass spectrometer with iFunnel for quantification (Agilent Technologies). Samples were diluted 10-fold into mobile phase A (see below), injected (10 μl) onto an Accucore C18 (150 × 2.1 mm, 2.6 μm particle size; Thermo Scientific), and separated using a 0.2 ml/min gradient of acetonitrile with 0.1% acetic acid (mobile phase B) in water with 0.1% acetic acid and 20 mM ammonium acetate (mobile phase A) as follows: 0–6 min: 3–20% B, 6.01–8 min: 100% B, followed by 4 min re-equilibration at 3% B.

Adduct quantification was performed by multiple reaction monitoring in the positive ionization mode (INH-NAD: 771 to 428, 771 to 136; ETH-NAD: 799 to 428, 779 to 136; PMD: 979 to 250, 979 to 632 and 979 to 136) at a collision energy (CE) of 30 and 55 V, respectively, with a cell accelerator voltage of 5, a dwell time of 100 ms and the following source settings: gas temperature: 200 °C, gas flow: 12 l/min, nebulizer: 15 psi, sheath gas temperature: 395 °C, sheath gas flow: 12 l/min, capillary voltage: 3000 V, nozzle voltage: 1500 V.

For putative identification of the adducts, pooled extracts from *Mtb* exposed to a drug at 10× MIC were concentrated twofold (ETH) or 10-fold (INH, PMD) by drying under vacuum (ETH: 60 μl dried in 30 min; INH, PMD: 600 μl dried in 3 h) without heating (Vacufuge Eppendorf), resuspending in mobile phase A, and centrifuging (5 min at 21,130 g). Fragmentation spectra were generated using a narrow quad isolation window of 1 amu and subject to CEs of 30 and 55 V. Generated spectra were compared to a synthesized standard (INH) or to predicted fragmentation spectra (ETH and PMD) using an in silico competitive fragmentation modeling algorithm[33]. All chemicals used for LC-MS/MS experiments were precured from Sigma, unless stated otherwise.

**Quantification and statistical analysis**. To determine statistical significance for TnSeq screens a nonparametric permutation test was used to calculate $P$ values. $Q$ values were obtained by adjusting for multiple comparisons using the Benjamin–Hochberg correction. Genes with a $Q$ value of <0.05 according to the permutation test were considered to be significant determinants of fitness in the presence of INH. TnSeq fold changes (TnSeq-FC) were computed as $\log_2$-transformed ratios of the normalized read counts between INH treated and untreated libraries[49]. All other statistical analyses were performed using GraphPad Prism 9.3.1 (GraphPad Software). The statistical tests used are indicated within figure legends and the corresponding $P$ values are reported as $*P < 0.05$, $**P < 0.01$, $***P < 0.001$, and $****P < 0.0001$.

**Reporting summary**. Further information on research design is available in the Nature Research Reporting Summary linked to this article.

## Data availability
Data generated during the study are available within the paper and/or its supplement. Source data are provided with this paper. We used the *Mtb* H37Rv reference genome (GenBank accession number NC_018143.1) to map transposon–chromosome junctions. Source data are provided with this paper.

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

## Acknowledgements

We thank Curtis Engelhart, Carolina Trujillo, Rodrigo Aguilera Olvera, Claire Healy, Heather Kim, Paula A. Pino Tamayo, Jennifer McConnell, Rosleine Antilus-Sainte, Anderson Watson, Katherine LoMauro, and Yan Pan for technical help. This work was funded by the NIH (Tri-Institutional TB Research Units U19AI111143 to D.S. and S.E., R25AI140472 to K.Y.R., P01AI095208 to J.C.S.), the Bill & Melinda Gates Foundation (OPP1177930 to K.Y.R), and the Welch Foundation (A-0015 to J.C.S.).

## Author contributions

K.M.K., R.S.J., T.E.H. A.G., R.W., I.V.K., J.P.S., M.X., M.D.Z., and M.G. performed the experiments. M.G., V.D., J.C.S., K.Y.R., D.S., and S.E. designed the experiments. K.M.K., D.S., and S.E. wrote the manuscript with input from the other authors.

## Competing interests

The authors declare no competing interests.
