## [Peer Review File · Nature Communications]

REVIEWER COMMENTS

Reviewer #1 (Remarks to the Author):

The manuscript of Kreuzfeldt et al., describes the contribution of CinA protein towards the development of drug tolerance in *Mycobacterium tuberculosis*. Genetic deletion of the *cinA* gene via homologous recombination sensitised *M. tuberculosis* against treatment with isoniazid, ethionamide, delamanid and pretomanid but not rifampicin. This was mainly due to the increased rate of killing rather than a reduction to the susceptibility of the delta-*cinA* strain to these drugs. The authors also eluded that this increase in bacterial killing is due to the fact that CinA can remove the NAD adduct from these drugs that is thought to be the active form of the drugs. This was determined using genetic studies of the knockout strain of CinA and complementing it with the different domains of CinA as well as using metabolomics where an increased accumulation of NAD adducts was observed in the delta-*cinA* strain. Lastly, the authors also argued that CinA could be a good target to shorten drug treatment duration in TB via limiting the amount of drug tolerant bacteria and increased rate of killing bypassing the generation of drug resistance. They demonstrated by the increased rate of killing using the drug combination of recent clinical trials, bedaquiline, pretomanid and linezolid in a murine model of infection. However, the authors did not address any potential effects that the CinA protein might have in the generation of resistance through enhanced tolerance neither the effect of overexpression of *cinA* in the wild type and possible effect on susceptibility or low-level resistance.

The work described in this manuscript is original and of high significance as it validates CinA as a potential target to achieve treatment shortening in TB. The methodology and data representation is solid but in Figure 2 and 5 instead of % survival, the absolute CFU values might have been more appropriate as it would indicate to what extent deletion or inhibition of CinA could enhance bacterial killing and if it approaches sterility levels.

Similarly, it would have been beneficial if the time kill curves were extended beyond day 4 to up to day 21 in vitro as again it would have been a good surrogate for overall killing and possibly address the generation of resistance or not, especially towards the later timepoints. This would also verify the effect of *cinA* towards bacterial tolerance or resistance, especially as the author did find out that deletion of *cinA* could contribute to a low-level resistance to pretomanid.

Specific comments:

Lines 93, 94 and 285-294: More details for the generation of the delta-cinA strain and the complementation strategy and plasmids used are required, and it would have been beneficial to the reader.

Lines 101-107: A table with MIC values (99 or absolute) and the rationale behind why the authors used the specified drug concentrations for their time kill kinetic experiments.

Lines 113-126: I think these data are very critical to support the effect of the CinA to increased killing via isoniazid. Figure 2 and S3 would have been more informative if they have also stated the absolute CFU and their reduction during INH treatment.

Lines 146-158: The genetic studies on the role of the pyrophosphatase domain requirement and the NAD adduct are well documented for all the mentioned drugs but not for pretomanid. Indeed, the implication of a bioactive pretomanid-NAD species is still hypothetical and the level of increased killing in the CinA KO not very strong compared to other drugs studied. Thus, I would have explored any link that the CinA might have with resistance instead of tolerance. A prolonged kill kinetic experiment would have added more value on this case and explore the potential link of CinA and resistance to pretomanid.

Lines 169-171: Metabolomics evidence also confirm a NAD accumulation mechanism for the bacterial tolerance seen via CinA. One extra condition to examine would have also been the overexpression of CinA in a wild type background and the effect on INH susceptibility and NAD product reduction.

Lines 182-184: Similarly, with above the plotting of absolute CFUs from mice experiments along with %survival would have been more informative.

Lines 207-209: This section seems to be confusing the link between persisters and tolerance that could then link to resistance as reference 8 suggests. Indeed, the subpopulation of persisters in the Delta-CinA strain might be lower (due to the increased killing) when compared to wild type; it's the reduced tolerant population that will minimize the generation of INH resistance.

Reviewer #2 (Remarks to the Author):

Isoniazid is commonly used in Mtb drug regimens, however, when Mtb enters its latent phase it is no longer susceptible to isoniazid treatment. In order to understand isoniazid resistance in Mtb the

authors undertook genetic screening of Mtb transposon mutants. Only one mutant, localized to *cinA*, was specifically susceptible to isoniazid treatment.

A Δ *cinA* Mtb was created and showed increased susceptibility to isoniazid in media growth experiments, inside macrophages, and in vivo. Interestingly, Δ *cinA* Mtb was not susceptible to rifampicin. In vivo, isoniazid treatment killed Δ *cinA* Mtb much more than wt and the mutant was more sensitive to lower drug doses than wt. Further, cultures of Δ *cinA* Mtb were killed faster than wildtype and showed ~60-fold fewer persister cells. To understand how *CinA* contributes to drug tolerance, inactive mutants were generated for its two functional domains - a pyrophosphatase domain and a PnnC deamidinase domain. Complementation of Δ *cinA* Mtb with domain mutants demonstrated that the pyrophosphatase domain is responsible for isoniazid tolerance.

The activity of isoniazid - as well as ethionamide and delamanid (and potentially pretomanid) – requires the formation of NAD-adducts, and notably, Δ *cinA* Mtb is more susceptible than wt to each of these therapeutics. Indeed, LC/MS-MS analysis of drug treated wt and Δ *cinA* Mtb showed increased accumulation of NAD-isoniazid and NAD-ethionamide adducts for Δ *cinA* Mtb. Complementation of Δ *cinA* Mtb with an active pyrophosphatase *CinA* mutant relieved the susceptibility of Δ *cinA* Mtb to the aforementioned drugs, and restored wildtype levels of adduct accumulation. Thus, *CinA* appears to cleave NAD-drug adducts via its pyrophosphatase domain, resulting in Mtb drug tolerance. Subsequently, the authors propose that drugs targeting *CinA* pyrophosphatase activity could improve the efficacy of Mtb drug regimens and help with the treatment of persister cells. Additionally, the authors checked for the efficacy of the BPaL drug cocktail on mice infected with wt and Δ *cinA* Mtb – and showed that Δ *cinA* Mtb was better targeted than wt Mtb, supporting the authors claim that targeting *CinA* could bolster current drug regimens.

Overall, this is a well-constructed and well-written paper. My only concern with this research carried out within lies in the mass spec discussion and analysis (Fig. 4). I wonder if the mass spec data are engaged with fully or presented as clearly as possible to the reader. While the research overall is well done, the manuscript would be greatly improved with mechanistic information for *CinA* and the NAD-drug adducts. In the current form, I am not sure the scope is ambitious enough for Nature communications.

Major comment:

1) The manuscript would be more impactful if the author demonstrated the *CinA* mechanism of action against NAD-drug adducts in vitro. First, soluble *CinA* from both *T. thermophilus* and *B. subtilis* has been previously recombinantly expressed and purified, and thus produced soluble Mtb *CinA* should not be a problem. Then the authors could show in vitro *CinA* binding experiments and/or *CinA* pyrophosphatase activity for NAD-drug adducts – or experiments could be just performed with the pyrophosphatase domain alone. A biochemical analysis would greatly improve the manuscript.

2) The mass spec analysis and discussion/display should be improved greatly. The reader is left feeling that the assumptions made from the MS data are to be believed as they are not carefully explained in the text and Figure 4 is difficult to navigate.

In Figure 4 – where is the information about standards? Could more conclusions be drawn from mass fragments in E/F?

Figure 4A/D

- can't see cinA knockout trace, perhaps results should not be displayed as overlay
- please define peak standards or show peak standards more clearly, and why was INH-NAD not run alone as a standard?
- Fig 4D colors should match 4A, the orange representing different things in both figures is not ideal
- What about the other peaks that accumulate in Figure 4A and 4D?

Figure 4 E/F could be better annotated

Minor comments:

1) A supplemental figure of the predicted structure of CinA and the position of the mutated residues would be much appreciated by readers.

2) These two are perhaps finicky concerns:

Fig 3A – DMD treatment, is the D80A mutant more susceptible to the drug than knockout? Is this likely?

Fig 3B – Delamanid and Pretomanid (and somewhat Isoniazid and ethionamide), the starting time point for K323A complement seems to have lower CFU – did this mutant show a growth defect?

3) Methods that are only used for supplemental figures should be moved to supplement – for example HPLC-MS/MS methodology for Fig S4, which appears to only be passingly referenced in the methods section.

We thank the editor and the reviewers for their positive feedback and helpful comments and critiques. We have addressed the specific points raised in their reviews. Below, please find the reviewers' comments, followed by our responses.

Reviewer #1 (Remarks to the Author):

The manuscript of Kreutzfeldt et al., describes the contribution of CinA protein towards the development of drug tolerance in *Mycobacterium tuberculosis*. Genetic deletion of the *cinA* gene via homologous recombination sensitised *M. tuberculosis* against treatment with isoniazid, ethionamide, delamanid and pretomanid but not rifampicin. This was mainly due to the increased rate of killing rather than a reduction to the susceptibility of the delta-*cinA* strain to these drugs. The authors also eluded that this increase in bacterial killing is due to the fact that CinA can remove the NAD adduct from these drugs that is thought to be the active form of the drugs. This was determined using genetic studies of the knockout strain of CinA and complementing it with the different domains of CinA as well as using metabolomics where an increased accumulation of NAD adducts was observed in the delta-*cinA* strain. Lastly, the authors also argued that CinA could be a good target to shorten drug treatment duration in TB via limiting the amount of drug tolerant bacteria and increased rate of killing bypassing the generation of drug resistance. They demonstrated by the increased rate of killing using the drug combination of recent clinical trials, bedaquiline, pretomanid and linezolid in a murine model of infection. However, the authors did not address any potential effects that the CinA protein might have in the generation of resistance through enhanced tolerance neither the effect of overexpression of *cinA* in the wild type and possible effect on susceptibility or low-level resistance.

The work described in this manuscript is original and of high significance as it validates CinA as a potential target to achieve treatment shortening in TB. The methodology and data representation is solid but in Figure 2 and 5 instead of % survival, the absolute CFU values might have been more appropriate as it would indicate to what extent deletion or inhibition of CinA could enhance bacterial killing and if it approaches sterility levels.

Similarly, it would have been beneficial if the time kill curves were extended beyond day 4 to up to day 21 *in vitro* as again it would have been a good surrogate for overall killing and possible address the generation of resistance or not, especially towards the later timepoints. This would also verify the effect of *cinA* towards bacterial tolerance or resistance, especially as the author did find out that deletion of *cinA* could contribute to a low-level resistance to pretomanid.

We thank the reviewer for acknowledging the originality and significance of the manuscript and the helpful suggestions. We have addressed all of them as described below.

Specific comments:

Lines 93, 94 and 285-294: More details for the generation of the delta-*cinA* strain and the complementation strategy and plasmids used are required, and it would have been beneficial to the reader.

Thank you for pointing this out. We have added more details regarding knockout construction and the complementation plasmids.

Lines 101-107: A table with MIC values (99 or absolute) and the rationale behind why the authors used the specified drug concentrations for their time kill kinetic experiments.

Thank you for raising this point. We added Supplementary Table 2 showing the MIC values of the tested drugs. For the kill curves we selected ~15x MIC for isoniazid and ethionamide because these concentrations were required to achieve substantial killing in wt Mtb. For delamanid and rifampicin 15x MIC did not result in significant killing, and we increased drug concentrations to achieve a ~10-fold reduction in CFU in 48 hrs. For pretomanid, we carried out new kill curves and adjusted the concentrations to 15x MIC. Because the MIC of pretomanid is 4-fold lower for $\Delta cinA$ than for wt, we performed kill curves with 2 pretomanid concentrations (4.5 $\mu\text{g/ml}$: 15 x MIC against wt and 1 $\mu\text{g/ml}$: 15x MIC against $\Delta cinA$). This revealed that the increased killing of the mutant was restricted to low drug concentrations. At 1 $\mu\text{g/ml}$ pretomanid, the wt did not get killed, while viability of $\Delta cinA$ declined. At 4.5 $\mu\text{g/ml}$ pretomanid both strains were killed with little difference in kill kinetics (see new Supplementary Fig. 4a). This is consistent with new data assessing the impact of isoniazid and pretomanid against Mtb cultured in a caseum mimetic (new Fig. 3b). Both drugs were more active against $\Delta cinA$, but this was concentration-dependent for pretomanid, which killed $\Delta cinA$ faster than wild type Mtb only at lower concentrations.

We don't know why the mutant is hypersusceptible to low but not high pretomanid concentrations but speculate that pretomanid might kill Mtb by multiple mechanisms, not all of which depend on the NAD adduct.

Lines 113-126: I think these data are very critical to support the effect of the CinA to increased killing via isoniazid. Figure 2 and S3 would have been more informative if they have also stated the absolute CFU and their reduction during INH treatment.

We have included absolute CFU data in the supplement (Supplementary Fig. 3e).

Lines 146-158: The genetic studies on the role of the pyrophosphatase domain requirement and the NAD adduct are well documented for all the mentioned drugs but not for pretomanid. Indeed, the implication of a bioactive pretomanid-NAD species is still hypothetical and the level of increased killing in the CinA KO not very strong compared to other drugs studied. Thus, I would have explored any linked that the CinA might have with resistance instead of tolerance. A prolonged kill kinetic experiment would have added more value on this case and explore the potential link of CinA and resistance to pretomanid.

We have performed prolonged kill kinetic experiments with pretomanid and isoniazid (new Supplementary Fig. 4a, b). They revealed that resistant mutants appeared in both wt and the CinA mutant. Since the CinA mutant was killed more rapidly before resistant mutants took over, the absolute number of resistant CinA mutant bacteria was smaller than the number of resistant wt bacteria at every time point. In addition, we were able to detect the pretomanid-NAD adduct and demonstrate that it accumulated in the CinA mutant (new Fig. 5d,e).

Lines 169-171: Metabolomics evidence also confirm a NAD accumulation mechanism for the bacterial tolerance seen via CinA. One extra condition to examine would have also been the overexpression of CinA in a wild type background and the effect on INH susceptibility and NAD product reduction.

Thank you for this suggestion. We constructed a CinA overexpressing strain (new Fig. 4a) and found that CinA overexpression increased INH tolerance (reduced killing by INH, new Fig. 4b), led to a 2-fold increase of the MIC (new Fig. 4c), and decreased INH-NAD adduct accumulation (new Supplementary Fig. 4e). However, owing to the decrease of INH-NAD levels below our limit of quantitation in the *atc* induced overexpressing strain, we are unable to accurately quantify the magnitude of this decrease and have included these data only in the Supplementary information.

Lines 182-184: Similarly, with above the plotting of absolute CFUs from mice experiments along with %survival would have been more informative.

CFU data have been added to the supplement (Supplementary Fig. 6a).

Lines 207-209: This section seems to be confusing the link between persisters and tolerance that could then link to resistance as reference 8 suggests. Indeed, the subpopulation of persisters in the Delta-CinA strain might be lower (due to the increased killing) when compared to wild type; it's the reduced tolerant population that will minimize the generation of INH resistance.

We agree with the reviewer that we cannot judge if and to what extent the findings reported by Levin-Reisman et al apply to $\Delta cinA$. We therefore removed this reference and revised this section of the discussion.

Reviewer #2 (Remarks to the Author):

Isoniazid is commonly used in Mtb drug regimens, however, when Mtb enters its latent phase it is no longer susceptible to isoniazid treatment. In order to understand isoniazid resistance in Mtb the authors undertook genetic screening of Mtb transposon mutants. Only one mutant, localized to *cinA*, was specifically susceptible to isoniazid treatment.

A $\Delta cinA$ Mtb was created and showed increased susceptibility to isoniazid in media growth experiments, inside macrophages, and in vivo. Interestingly, $\Delta cinA$ Mtb was not susceptible to rifampicin. In vivo, isoniazid treatment killed $\Delta cinA$ Mtb much more than wt and the mutant was more sensitive to lower drug doses than wt. Further, cultures of $\Delta cinA$ Mtb were killed faster than wildtype and showed ~60-fold fewer persister cells. To understand how CinA contributes to drug tolerance, inactive mutants were generated for its two functional domains - a pyrophosphatase domain and a PnnC deamidinase domain. Complementation of $\Delta cinA$ Mtb with domain mutants demonstrated that the pyrophosphatase domain is responsible for isoniazid tolerance.

The activity of isoniazid - as well as ethionamide and delamanid (and potentially pretomanid) – requires the formation of NAD-adducts, and notably, $\Delta cinA$ Mtb is more susceptible than wt to each of these therapeutics. Indeed, LC/MS-MS analysis of drug treated wt and $\Delta cinA$ Mtb showed increased accumulation of NAD-isoniazid and NAD-ethionamide adducts for $\Delta cinA$ Mtb. Complementation of $\Delta cinA$ Mtb with an active pyrophosphatase CinA mutant relieved the susceptibility of $\Delta cinA$ Mtb to the aforementioned drugs, and restored wildtype levels of adduct accumulation. Thus, CinA appears to cleave NAD-drug adducts via its pyrophosphatase domain, resulting in Mtb drug tolerance. Subsequently, the authors propose that drugs targeting CinA pyrophosphatase activity could improve the efficacy of Mtb drug regimens and help with the treatment of persister cells. Additionally, the authors checked for the efficacy of the BPaL drug cocktail on mice infected with wt and $\Delta cinA$ Mtb – and showed that $\Delta cinA$ Mtb was better targeted than wt Mtb, supporting the authors claim that targeting CinA could bolster current drug regimens.

Overall, this is a well-constructed and well-written paper. My only concern with this research carried out within lies in the mass spec discussion and analysis (Fig. 4). I wonder if the mass spec data are engaged with fully or presented as clearly as possible to the reader. While the research overall is well done, the manuscript would be greatly improved with mechanistic information for CinA and the NAD-drug adducts. In the current form, I am not sure the scope is

ambitious enough for Nature communications.

Major comment:

1) The manuscript would be more impactful if the author demonstrated the CinA mechanism of action against NAD-drug adducts in vitro. First, soluble CinA from both *T. thermophilus* and *B. subtilis* has been previously recombinantly expressed and purified, and thus produced soluble Mtb CinA should not be a problem. Then the authors could show in vitro CinA binding experiments and/or CinA pyrophosphatase activity for NAD-drug adducts – or experiments could be just performed with the pyrophosphatase domain alone. A biochemical analysis would greatly improve the manuscript.

We agree that demonstrating CinA's activity biochemically would support our claims. We devoted significant effort to accomplishing this. Unfortunately, expression of Mtb's CinA in *E. coli* resulted in insoluble protein, and an attempt to overexpress CinA in *M. smegmatis* did not yield sufficient protein amounts for purification. Overexpression of only the pyrophosphatase domain also produced insoluble protein. We were able to refold the insoluble proteins to some extent, which was only possible in high concentration of phosphate-citrate buffer. The refolded protein did, however, not show any pyrophosphatase activity on the model substrate ADP-ribose or INH-NAD, despite testing several divalent cations (Cialabrini et al., 2013) and using LC-MS detection. We additionally tested activity using the AMP-Glo kit but found that the phosphate-citrate buffer was incompatible with this assay and that, unfortunately, proteins precipitated again while diluted into all other assay buffers tested. Refolded protein was useful to produce the antibodies but proved to be insufficiently folded/soluble/active for the activity assay. For these reasons we were unable to characterize CinA or its pyrophosphatase domain biochemically.

2) The mass spec analysis and discussion/display should be improved greatly. The reader is left feeling that the assumptions made from the MS data are to be believed as they are not carefully explained in the text and Figure 4 is difficult to navigate.

In Figure 4 – where is the information about standards? Could more conclusions be drawn from mass fragments in E/F?

Figure 4A/D

- can't see cinA knockout trace, perhaps results should not be displayed as overlay
- please define peak standards or show peak standards more clearly, and why was INH-NAD not run alone as a standard?
- Fig 4D colors should match 4A, the orange representing different things in both figures is not ideal
- What about the other peaks that accumulate in Figure 4A and 4D?

Figure 4 E/F could be better annotated

We performed additional experiments to more convincingly demonstrate the identity of the INH-NAD adduct by high- instead of low-resolution fragmentation spectra. These high-resolution spectra allow high-confidence annotations and demonstrate a similar pattern of fragmentation to a chemical standard with annotation of key NAD-derived fragments, as well as annotation of similar NAD-derived fragments in the corresponding ETH- and pretomanid-NAD adducts. These are now included in a significantly revised Figure 5 and Supplementary Fig. 5 that are more clearly and consistently annotated. We confirmed the presence of a CinA- and INH-dependent INH-NAD adduct in Mtb lysates using an INH-NAD standard that was synthesized by chemical oxidation of INH with stoichiometric amounts of manganese pyrophosphate and NAD(H) (PMID: 11948876, 12069966). The identity of the synthesized standard was confirmed by annotation of the MS/MS fragmentation spectra corresponding to its predicted exact mass and structure and

matched those of a metabolite with the same LC retention time in INH-treated samples. These data are now shown in Figure 5a.

Minor comments:

1) A supplemental figure of the predicted structure of CinA and the position of the mutated residues would be much appreciated by readers.

Thank you for this suggestion. We added a supplemental figure of the predicted CinA structure with indication of the mutated residues (Supplementary Fig. 2b).

2) These two are perhaps finicky concerns:

Fig 3A – DMD treatment, is the D80A mutant more susceptible to the drug than knockout? Is this likely?

Fig 3B – Delamanid and Pretomanid (and somewhat Isoniazid and ethionamide), the starting time point for K323A complement seems to have lower CFU – did this mutant show a growth defect?

The increased susceptibility to delamanid of the D80A mutant compared to the knockout was reproducible in 2 independent experiments, but only observed at the higher drug concentration. In all other assays the knockout and the D80A mutant phenocopied each other.

The reviewer is correct all CinA mutants have slight growth defects (see figure below and Supplementary Fig. 4c), and the K323A complement strain is also clumpier than the other strains, which sometimes resulted in a reduced starting inoculum.

3) Methods that are only used for supplemental figures should be moved to supplement – for example HPLC-MS/MS methodology for Fig S4, which appears to only be passingly referenced in the methods section.

We moved methods that were only used for supplemental figures to the Supplementary Information file.

REVIEWERS' COMMENTS

Reviewer #1 (Remarks to the Author):

The present manuscript describes the role of CinA in multidrug tolerance in *M. tuberculosis*. These results are very relevant in understanding the role of CinA in bacterial tolerance and its potential to decrease the treatment period of tuberculosis. The authors clarified the limitations of this research on the biochemical characterisation of the CinA as well as they were able to demonstrate the existence of the pretomanid-NAD adduct and eluded towards the interesting finding that the amount of pretomanid varies the response of the mutant strain. The work in this revised form is original, innovative and of high significance and an important contribution in the TB field.

Reviewer #2 (Remarks to the Author):

As the manuscript has already been reviewed positively, my comments are on the revisions only.

The authors have addressed all the reviewers comments completely, and the manuscript is greatly strengthened through their efforts. This study should be published in Nature Communications.